# Development and Preliminary Evaluation of the Effects of a Preceptor Reflective Practice Program: A Mixed-Method Research

**DOI:** 10.3390/ijerph192113755

**Published:** 2022-10-22

**Authors:** Heui-Seon Kim, Hye-Won Jeong, Deok Ju, Jung-A Lee, Shin-Hye Ahn

**Affiliations:** Department of Nursing, Chonnam National University Hospital, Gwangju 61469, Korea

**Keywords:** preceptorship, nurses, emotional intelligence, program evaluation, nursing education research, reflective practice

## Abstract

Studies on methods to share nursing and learning experiences with preceptors are lacking. This study was conducted to determine the preliminary effects of developing and applying a reflective practice program for preceptor nurses who experience stressful situations to convert negative emotions into positive ones. This study was conducted over 12 weeks from March to May 2022 on 47 participant nurses in South Korea. Preceptor Reflective Practice Program (PRPP) was conducted in parallel with writing a reflective journal and a reflective practice workshop. Data collection was integrated through quantitative and qualitative approaches. Quantitative data were collected through questionnaires on stress coping, the burden of preceptors, social support, and emotional intelligence, and analyzed by SPSS WIN 26.0 program. The questionnaire data were analyzed after the preceptor nurses had written a reflective journal. Stress coping, social support, and emotional intelligence significantly increased in preceptor nurses after participating in the PRPP. This study found that the PRPP helped nurses improve their emotional intelligence through reflective practice and convert stress into a more positive direction. Therefore, at the organizational and national levels, a reasonable compensation system to provide support workforce and to the work of preceptor nurses should be established.

## 1. Introduction

In order to provide quality nursing care to patients in a rapidly changing healthcare environment, it is important to train excellent nursing personnel. In particular, new nurses must develop their competence by training in systematic education programs [1]. The preceptorship system is widely used in many medical institutions to provide new nurses with clinical field adaptation and training, and to bridge the gap between theory and practice [2]. Preceptor nurses perform nursing tasks with new nurses, simultaneously train new nurses, and promote adaptation and organizational socialization of new nurses in nursing unit [3]. The role of preceptor nurses in helping new nurses adapt to their work and organization is becoming more important in hospitals with a high turnover rate of new nurses [4]. Positive preceptor nurses may promote organizational socialization by helping new nurses improve their problem-solving abilities in clinical practice, strengthening positive emotions, and enhancing resilience [5]. Preceptor nurses should have excellent communication skills and provide feedback, and not solely act as a conduit or channel of education in order to provide effective education to new nurses [4].

The average clinical experience of preceptor nurses is approximately 3 years, and they often work as preceptors before becoming competent as nurses [6]. Preceptor nurses are often overloaded with work and fail to sufficiently educate new nurses because they are required to perform their daily nursing tasks simultaneously [7]. It was previously reported that the educational effect on new nurses decreased and negatively affected new and preceptor nurses themselves if the preceptor nurses did not have the appropriate competency [4]. In addition, preceptor nurses often take on the role of preceptors before acquiring the required competency level [7]. Preceptor nurses feel burdened and stressed in the atmosphere of a nursing organization that appoints the preceptor responsible for all the tasks and responsibilities related to the job of a new nurse [8] and, thus, tend to avoid the role of the preceptor [9]. In addition to other healthcare workers, preceptor nurses are the first to encounter new nurses, but because of the lack of clinical experience of new nurses, negative communication, such as reprimands, irritation, neglect, and authoritative teaching methods, are common among preceptor nurses [10]. There are also reports of cases where the preceptor was hard because of the perspective of colleagues who think that the preceptor taught a new nurse incorrectly when a patient safety accident occurred due to a mistake or error committed by a new nurse in the department [1,8]. Both the demand and stress associated with the preceptor’s role for nurses increased. If the preceptorship education period increases to more than 8 weeks, a further burden is placed on the preceptor nurse [11]. As a result of cumulative fatigue from work aggravation, the nurse was exposed to an unbearable situation of having to train a new nurse, resulting in mental exhaustion [1]. Further, nurses who experience such stress no longer find pleasure in work, the quality of patient care deteriorates, and they become dissatisfied with the nursing profession and eventually quit work [12]. Therefore, it is essential to provide high-quality coping mechanisms for nurses to manage work stress [13].

Stress is a part of daily life, and it may be impossible to evade it completely. This depends on how one experiences and copes with stress, making it important to learn new stress coping methods [14]. Diverse stress coping strategies, including problem-solving skills, social support, and positive thinking, may improve nurses’ job commitment and satisfaction, thus reducing the turnover rate [14]. Emotional intelligence is the ability to understand, empathize, organize, and control the emotions of oneself and others [15]. The higher the emotional intelligence, the greater the number of positive emotions that can be used in stressful situations to cope with stress effectively [16]. Furthermore, as emotional intelligence may be improved through training or education [17], it is necessary to develop such programs. 

Reflective practice is a process of critically evaluating one’s work performance by critical thinking to adjust practice in a desirable manner [18]. New understanding and insights can be gained through the process of reflecting on one’s clinical experience using narration in the context of reflective practice [19]. Nurses who gain insight through reflective practice can understand their behavior, respond appropriately to clinical issues, and handle threatening factors, such as anxiety and stress, better [19]. Reflective practices, such as reflective journaling, may effectively help deal with negative emotions and stress, and fully understand and change one’s emotions into positive ones by sharing experiences with colleagues [20]. There are verbal methods, written methods, or a combination of the two strategies for reflective practice [21], with most studies focusing on written methods [20]. Conversation, which is a verbal method for reflective practice, is essential and allows us to deal more deeply with reflections made through writing and gain broader insight [19]. Nurses have the advantage of feeling emotional stability and improving personal insight through insights of other people by recognizing their problems through reflective practices at the group level [22]. Educational strategies, such as role-playing, films, and videos for preceptors’ reflection practice training, are helpful for improving the relationship between preceptors and new nurses; however, they are rarely used in preceptor education [23]. Moreover, there are no studies on applying verbal or combination methods to share nursing and learning experiences with preceptors or colleagues through workshops conducted by facilitators or to discuss and reflect on issues raised [20].

Studies that have been conducted previously on preceptor nurses mainly focused on the experience of preceptor nurses [1,24], teaching styles [25], and communication types [26]. In a literature review study on occupational stress and mental health intervention programs [27], it was reported that domestic studies were predominantly biased on the prevention and support of new nurse turnover and that there were few studies on tenured nurses. As for intervention research for preceptor nurses, an action study developed a preceptor education program based on the one-minute preceptorship model [9], and it suggested the development of a program for the emotional management of preceptor nurses. Therefore, by developing and applying a reflective practice program for preceptor nurses who experience various stressful situations, we intend to examine the preliminary effect on reducing work burden and shifting emotions in a positive direction.

## 2. Materials and Methods

### 2.1. Theoretical Framework 

The theoretical framework of the Preceptor Reflective Practice Program (PRPP) was based on Gibb’s reflective cycle [28] (Figure 1). This study was used to structure the reflective practice of preceptor nurses through Gibb’s model. This model consists of description, feelings, evaluation, analysis, conclusion, and action plan, and is widely used for the development of reflection skills. In this study, reflective thinking was induced by expressing one’s thoughts on background information, using verbal and written methods for the experience of the preceptor in the description stage. Subsequently, in the feeling state, the preceptor nurses discussed their feelings and thoughts on their own experience and could speak frankly concerning their feelings. Next, in the evaluation state, opinions were collected via discussion with fellow nurses and clinical educator nurses. Further, in the analysis state, the preceptor nurses’ own reflection could be related to other people’s opinions and experiences. Subsequently, in the conclusion state, preceptor nurses distinguish and recognize what they can and cannot to do change their situation. Finally, in the action plan state, the upcoming activities were reflected by summarizing the matters that the preceptor should improve and perform.

### 2.2. Study Design

A mixed research design was used to develop and apply the PRPP and to evaluate the preliminary effect produced. To assess the effectiveness of PRPP, a one-group pretest-post-test design was utilized. Qualitative research was conducted by adopting a generic qualitative research design to explore participants’ views and experiences to understand the phenomenon in depth and to help researchers articulate participants’ perspectives [29,30]. The contents of self-reflective journals prepared by preceptor nurses who participated in the program were analyzed to explore their experience and reflective practice process during the preceptorship. 

### 2.3. Participants

The participants of this study were preceptor nurses working in a tertiary hospital in South Korea and were undergoing preceptorship from March to May 2022. Preceptor participation conditions included nurses who listened to the explanation of this study, understood the purpose of the study, voluntarily agreed to participate and had a total clinical experience of >3 years. Among the nurses who played the role of a preceptor nurse when participating in this study, those who voluntarily agreed to participate in the study were targeted. All preceptor nurses expressed their consent to participate in the study and participated in the study as a single group. The participants were provided with 2 days of educational leave when they participated in two reflective practice workshops. Therefore, there was no disadvantage in their work. The sample size was calculated using G*Power 3.1.9.4 (Universitat Dusseldorf, Dusseldorf, Germany) (significance level (α = 0.05), effect size (d = 0.46), and power (1 − ß = 0.90)), and the effect size of Lee et al. [31] was referenced. In our work, the minimum number of participants required was 42; considering a dropout rate of 10%, 47 individuals were enrolled. 

### 2.4. Research Procedure

#### 2.4.1. Development of the PRPP

The PRPP aims to reduce the work burden of preceptor nurses and change their emotions in a positive direction. The PRPP comprised a combination of written and verbal strategies based on the results of the prior literature review for reflective practice [21] and was employed over 12 weeks during preceptorship (Figure 2). The content validity of this program was checked by two nursing professors, one nursing education team leader, and three clinical nurse educators with a master’s degree in nursing; the Content Validity Index (CVI) was 0.94. Writing is one of the cognitive techniques that help individuals discover life patterns that have a negative impact. Moreover, it helps them experience catharsis through writing by allowing them to look back and analyze events in their life, environment, and their emotions [32]. Therefore, in this program, a self-reflective journal written once every 3 weeks, for 12 weeks, through Padlet, an online platform, resulting in four entries was composed for preceptor nurses for written reflective practice. The self-reflective journal presented a reflective guide question to enable critical inquiry based on previous research [33] (e.g., “What happened?”, “How did this make you feel?”, “How did you feel while training a new nurse?”, “What were the important background factors that led to this experience?”, “How have past experiences helped you understand your current situation?”, “Moreover, what kind of help or effort do you need to have a more positive experience when you are put in a similar situation again in the future?”). Clinical nurse educators provided social support by writing emotional and supportive comments in self-reflective journals. The verbal reflective practice workshop took place twice for 8 h per session, over 1 and 3 months of preceptorship, for a total of 16 h. A facilitator conducted the workshop, and the first session allocated time for sharing experiences and empathizing with fellow preceptors through group discussion. In addition, discussions were conducted to strengthen positive emotions. A role-play strategy for improving positive communication and relationships was used to help shift the negative communication experienced in preceptorship into a more effective one. Role-playing corresponds to roles assigned to various situations that can occur in real situations, thus, providing opportunities to think more deeply while writing scenarios with learning activities that speak and act according to the situation. Moreover, it provides understanding and empathy for the situation [34]. The communication role-play was made using a video demonstration method of the communication scenario selected through group discussion. A scenario was written that required nurses to share difficult communication cases with new nurses or follow nurses in their group, and to choose one case for a role-playing exercise. In the second session, growth strategies through preceptor experience were conceived through group discussions, and opinions were shared with fellow preceptors on stress management methods. Using a video demonstration, the results derived through group discussion were used to create a video on this growth strategy through preceptor experience. After the group discussion, the training and preceptor nurses listened to preceptor nurses’ difficulties, gave advice, and mentored them. Finally, preceptor nurses had time to share their impressions of participating in the program. A summary of the PRPP activities of preceptor nurses is presented in Table 1. Written reflective practice activities using an online platform and photos of growth strategies through experiences and sharing experiences as part of verbal reflective practice activities were presented (Table 1).

#### 2.4.2. Application of the PRPP

Data collection was conducted from 2 March 2022 to 25 May 2022. The preliminary survey was conducted before the initiation of the PRPP, and the follow-up survey was performed immediately after the end of the PRPP. Qualitative research data were collected for 12 weeks (2 March 2022 to 25 May 2022) from the initiation to the end of the program. The self-reflective journal was written using structured reflection questions. When writing the self-reflective journal, a personal ID was given, and for qualitative content analysis, the journal contents were transcribed to Microsoft Office Excel (Microsoft Inc., Redmond, WA, USA).

#### 2.4.3. Ethical Consideration

Data collection was initiated after obtaining approval from the Institutional Review Board of the Chonnam National University Hospital (IRB No: CNUH-2022-247). After explaining the purpose and method of the survey to the study participants, they received consent for participation in the study, and were explained that they could withdraw at any time if they did not want to participate. It was explained that the collected data will be used only for research purposes and will be treated as anonymous. The collected data were stored in a secure place by giving only a personal ID, and the survey data were coded and used only for research purposes to ensure confidentiality and anonymity. After the end of the program, a small token of appreciation was given to the study participants.

### 2.5. Data Collection and Analysis

#### 2.5.1. Data Collection

The participants’ general characteristics consisted of age, sex, marital status, total clinical experience, department, spontaneity in the role of preceptor nurse, number of preceptor experiences, and willingness to continue the role of preceptor nurse.

In this study, we used the Way of Stress Coping Checklist, developed by Lazarus and Folkman [35], adapted by Kim and Lee [36], and measured by Park [37], as a modified and supplemental tool. This tool comprises a total of 24 items, and each question consists of 12 items on active coping (i.e., six items for problem-focused coping and six items for seeking social support coping), and 12 items on passive coping strategies (i.e., six items for emotional focusing stress coping and six items for desire thought). Responses were measured using a 4-point Likert scale ranging from 1 (not used) to 4 (very used) points. A higher score indicated that more coping methods of the corresponding area were used. Cronbach’s α was 0.81 in Park’s study [37] and 0.70 in our work.

The burden on preceptor nurses was measured with a tool developed by Park et al. [38]. This tool consists of 25 items. The sub-domain comprises eight items of burden related to new nurses, 13 items of work burden related to oneself, and four items related to colleagues and others. Responses were measured using a 5-point Likert scale ranging from 0 (rarely) to 4 (very much) points; the higher the score, the higher the burden. At the time of development, the burden related to new nurses was Cronbach’s α = 0.65, the work burden related to oneself was 0.80, and the burden related to colleagues and others was 0.64. In this study, the burden related to new nurses was Cronbach’s α = 0.76, while the work burden related to oneself was 0.85. The burden related to colleagues and others was 0.52.

To measure the social support of clinical nurse educators, the authors modified and supplemented Park’s social support tool [39]. The content validity of this tool was checked by two nursing professors, one nursing education team leader, and three clinical nurse educators with a master’s degree in nursing, and the CVI was 0.95. This tool has a total of 20 items. Responses were measured using a 4-point Likert scale ranging from 1 (not at all) to 4 (strongly agree) points; the higher the score, the higher the perceived social support by clinical nurse educators. In the study by Choi and Yoo [40], which investigated social support for insurance review nurses, Cronbach’s α was 0.96, while in this study, it was also 0.96.

For emotional intelligence, we used the version of Wong and Law’s Emotional Intelligence Scale (WLEIS) [17] that was translated into Korean by Yang and Kang [41]. WLEIS has a total of 16 items, and the sub-domain consists of four items of self-emotions appraisal, four items of others-emotions appraisal, four items concerning regular emotion, and four items on the use of emotion. Responses were measured using a 5-point Likert scale ranging from 1 (not at all) to 5 (strongly agree) points; the higher the score, the higher the emotional intelligence. In the study by Yang and Kang [41], Cronbach’s α was 0.88, while in this study, Cronbach’s α was 0.92.

#### 2.5.2. Data Analysis

Quantitative data were analyzed using the SPSS/WIN 26.0 (IBM Corp., Armonk, NY, USA) program. The participants’ general characteristics data, stress coping styles, the burden of preceptors, social support, and emotional intelligence were normally distributed and were analyzed with descriptive statistics. Owing to the small sample size in this study, the normality assumption was tested with the Shapiro–Wilk test [42]. To analyze the effect of the intervention, the difference between the pre- and post-test scores of dependent variables, including stress coping styles, burden, social support, and emotional intelligence, was analyzed by paired t-test or Wilcoxon signed-rank test. The reliability of the measured variables was calculated using Cronbach’s alpha coefficient.

The self-reflective journal, which consists of qualitative data, was downloaded through the Excel program. As an online platform was used to prepare the reflection log, the researcher went through a proofreading procedure for misspellings and compound words. In cases where the content was confusing, the meaning was made clear by reconfirming with the author. Qualitative content analysis [43] was applied as a data analysis method to reveal patterns and topics of content through the coding process, which is a systematic classification method based on holistic understanding of data. Qualitative data were managed using NVivo version 12 (QSR International, Burlington, MA, USA). The specific analysis procedure is as follows. First, two independent researchers (HW Jeong and SH Ahn) tried to improve the overall understanding of the preceptor nurse’s preceptorship experience by repeatedly reading the entire reflection journal. Second, to discover the main code by continuously reading the contents several times, meaningful sentences containing core thoughts and concepts were extracted and coded. Third, we categorized the extracted meaningful sentences by interconnecting them to make them more abstract. Fourth, based on the meaning and relevance of these categories, the task of naming the categories was repeated so that the interrelationship could be understood. Finally, based on the final analysis results, an in-depth description of each category was attempted, and the reliability of the categories was verified by returning to the original data centering on these categories and reading and analyzing them as a whole. The analyzed results were reviewed for validity by one physician with experience in conducting qualitative research.

### 2.6. Rigor

One author with a Ph.D. degree with experience in qualitative research and one doctoral student co-author took qualitative research classes, shared data, and participated in the analysis process to crosscheck their work. To secure rigor, the qualitative data analysis was performed according to the four criteria—true value, applicability, consistency, and neutrality—proposed by Lincoln and Guba [44]. First, the preceptor nurses were allowed to express their thoughts to secure true value freely. Then, we categorized and analyzed the self-reflective journals, showed the results to the three participants, and confirmed that the derived themes and meanings were accurately conveyed. Second, preceptor nurses were required to fill out self-reflective journals to ensure applicability, and the data was completed at the program’s end. Third, to secure consistency, the data collection and analysis process was described in detail by two authors who were experienced in qualitative research and took qualitative research classes, who continuously discussed and evaluated the research process and results. Finally, the author consciously tried to exclude prejudices to secure neutrality. To maintain the reliability of the data analysis results, the analysis was repeated until consent was reached with the co-authors. Finally, the data were repeatedly compared and analyzed to increase the sensitivity of the analysis.

## 3. Results

### 3.1. General Characteristics of the Participants

The general characteristics of the participants of this study are presented in Table 2. The average age of the participants was 27.81 years, 53.2% of whom were aged between 27 to 29 years, and 93.6% were women. A total of 93.6% of the preceptor nurses were single. The average clinical career length was 4.85 years, and most of the participants were nurses for a period < 6 years (85.2%). The most common work unit was the surgical wards, with 21.3% of the nurses working there. A total of 19.1% of the participants voluntarily applied for the preceptor role, 53.2% answered that it was their first time working as a preceptor, and 51.1% answered that they wanted to become a preceptor next time.

### 3.2. Preliminary Effects of the Preceptor Reflective Practice Program

The results of applying to the PRPP are presented in Table 3. The participants’ ability to cope with stress significantly increased from 2.80 (±0.25) to 2.89 (±0.30) after the program. Examining each sub-domain, passive coping significantly increased from 2.59 (±0.29) to 2.71 (±0.34) after the program, while the emotional focusing stress coping significantly increased from 2.32 (±0.40) to 2.49 (±0.47). Social support significantly increased from 3.23 (±0.42) to 3.36 (±0.46), and emotional intelligence significantly increased from 3.52 (±0.63) to 3.82 (±0.58). Examining each sub-domain of emotional intelligence, others-emotion appraisal significantly increased from 3.79 (±0.73) to 4.04 (±0.65), regulation of emotion significantly increased from 3.34 (±0.84) to 3.69 (±0.71), and use of emotion significantly increased from 3.34 (±0.84) to 3.69 (±0.71). However, there were no significant differences in the categories of active coping, the burden of preceptors, and self-emotions before and after the program.

### 3.3. Content Analysis of Self-Reflective Journals

During the 8-week period of the PRPP, the preceptor nurses wrote a self-reflective journal on four different occasions. The participants in the program shared their experience in, and feelings towards, training new nurses with fellow preceptors, gaining empathy and support through comments from clinical nurse educators, and the shared problem-solving methods. The results of the analysis of the preceptor nurses’ self-reflective journals are as follows.

#### 3.3.1. Theme 1. Disappointing Preceptor Education Support System

Preceptor nurses felt that the hospital-level support was insufficient while educating new nurses, and a shortage of workforce and work overload were experienced. Preceptor nurses felt it burdensome and difficult to train two or more new nurses simultaneously and train new nurses while working together.

(1)Insufficient educational support system



*“It is very hard for one preceptor to train two new nurses. It was necessary to check whether each individual understood the contents of the training, and it was too difficult to explain several times because each new nurse had a different acquisition rate”.*

*(Participant 3, Participant 12, Participant 14, Participant 19, Participant 32)*





*“The greatest difficulty in training two new nurses is that one of them follows, but the other one often does not know about it, and it was difficult to train because of the different degrees of activeness. When I saw a new nurse who could not follow, I was very stressed because I was compared to a new nurse who was good at it”.*

*(Participant 20, Participant 37)*





*“As I teach two new nurses at once, I think that the new nurse’s growth rate seems to be slow as the new nurses have to divide the work they have to do alone”.*

*(Participant 9)*





*“New nurses need to learn several things, but it feels increasingly burdensome because two people cannot be trained simultaneously”.*

*(Participant 12)*



(2)Parallel work and education



*“I know that it is inevitable owing to the coronavirus disease 2019 (COVID-19) pandemic, but there are no training days for new nurse training; therefore, I had to train new nurses while caring for patients, and so it was very burdensome and daunting”.*

*(Participant 5, Participant 7, Participant 11, Participant 20, Participant 21, Participant 32, Participant 42, Participant 46)*





*“There are too many things to teach new nurses, but the situation is not supported. Hence, I feel frustrated, impatient, and anxious”.*

*(Participant 9, Participant 19, Participant 35, Participant 42)*





*“As I was teaching new nurses while also caring for patients, the flow of education was interrupted, and there were many days when I only gave half of the information and could not tell the whole thing. Thus, I think it was really stressful at work”.*

*(Participant 5, Participant 13, Participant 20)*



#### 3.3.2. Theme 2. Complex Emotional Expression of the Preceptor Role

Preceptor nurses who acted as preceptors for the first time first expressed their excitement. However, the team leader suddenly notified most that they had to train new nurses, thereby placing the burden of training new nurses on preceptor nurses without adequately preparing them for their teaching role. The preceptors felt that they had a duty to teach new nurses well, but felt that they lacked the ability to do so. As the preceptor nurses were training the new nurses, they felt that the time was flying by more quickly than expected. Moreover, they felt pressured by the few remaining new nurses to become independent as soon as possible. In addition, the preceptor nurses were worried about whether the new nurses would be able to work independently, but also felt impatient to teach more, and felt bittersweet at the end of the preceptorship process.

(1)Excitement regarding the first-time preceptor role



*“Being a preceptor for the first time, I am nervous and excited, and I want to gather the strengths of good preceptors and teach them as best as I can”.*

*(Participant 19, Participant 40, Participant 46)*





*“Having been a new nurse at one time in the past, I know that a new nurse really wants to meet a preceptor, so I also want to study and prepare hard to teach”.*

*(Participant 19, Participant 46, Participant 50)*





*“I want to be a preceptor who can kindly and generously answer and educate new nurses when they ask a question and approach them like a friend”.*

*(Participant 46, Participant 49, Participant 50)*



(2)The burden of the preceptor role



*“As my career progressed, the burden of becoming a preceptor increased, and I received a notice from the team leader to become a preceptor. I am scared and worried about whether I will do well”.*

*(Participant 17, Participant 18, Participant 20, Participant 22, Participant 24, Participant 32, Participant 41, Participant 42)*





*“When new nurses do not work well in the department, I think that new nurses do not adapt well because I teach incorrectly, and it is difficult and burdensome to train new nurses”.*

*(Participant 4, Participant 14, Participant 28)*





*“There are many things I do not know yet, but the burden and pressure to train new nurses was too great”.*

*(Participant 8, Participant 29, Participant 32, Participant 46)*



(3)Pressure because of the limited time



*“I have not much to tell new nurses, but time flies so fast that I still have a lot to tell them, but I am worried whether I can tell them all before they become independent”.*

*(Participant 2, Participant 3, Participant 10, Participant 13, Participant 28, Participant 33)*





*“There are many things I want to tell new nurses, but as independence approached, I felt impatient as if I were about to become independent, so I scolded and rushed the new nurses”.*

*(Participant 13, Participant 14, Participant 18, Participant 28, Participant 29)*





*“When new nurses become independent, they will have to deal with the work alone and should be of some help to the ward, but they are still clumsy in their work, so I feel heavy and worried ahead”.*

*(Participant 7, Participant 14, Participant 18, Participant 19, Participant 20, Participant 24)*





*“I feel bittersweet because I have become attached to new nurses”.*

*(Participant 10, Participant 19, Participant 20, Participant 32, Participant 33, Participant 42)*



#### 3.3.3. Theme 3. Self-Growth through the Preceptor Role

Preceptor nurses felt a lack of competence; however, they wanted to study and systematically train new nurses, but were disappointed that it did not go as well as planned. Preceptor nurses felt reward and growth as they trained the new nurses. They also reposted, recalling their early nursing days. The preceptor nurses were determined to train the new nurses better.

(1)Feeling a lack of competence as a preceptor



*“There were many days when I felt that I needed to study more because I was afraid that I would give wrong information while training new nurses, and I felt a lot of things and reflected on myself watching new nurses learning while following me”.*

*(Participant 7, Participant 10, Participant 13, Participant 40)*





*“Even though I made a plan to teach new nurses, I was upset because it did not go as expected owing to the nature of the ward”.*

*(Participant 3, Participant 29, Participant 43)*





*“I am so sorry and reflect on myself because I am not good enough to wait for new nurses, and I am just scolding them and not giving them a good education”.*

*(Participant 14, Participant 24, Participant 40, Participant 43)*



(2)Commitment as a preceptor



*“Although the new nurses are still clumsy and slow, I should support and encourage them so that they may adapt well to the ward and grow well after becoming independent”.*

*(Participant 8, Participant 9, Participant 16, Participant 17, Participant 18, Participant 19)*





*“When I was a new nurse, I was full of things I did not know, but I should remember that time and teach new nurses”.*

*(Participant 2, Participant 7, Participant 12, Participant 20, Participant 24, Participant 26, Participant 39)*



(3)Personal growth as a preceptor



*“It is meaningful that I am having a new experience as a preceptor, and I hope that new nurses will grow well”.*

*(Participant 3)*





*“In order to teach new nurses, it was an opportunity to redefine the knowledge I had learnt and known, and it was an opportunity for growth”.*

*(Participant 5, Participant 8, Participant 13, Participant 18, Participant 39)*



## 4. Discussion

This study attempted to determine the preliminary effects of the PRPP on preceptor nurses’ stress coping strategies and burden, social support, and emotional intelligence, as well as the reflective practice process through content analysis of a self-reflective journal.

First, the PRPP in this study was found to have a significant effect on increasing emotional focusing as a stress coping among the preceptor nurses; however, it did not increase active coping. This was contrary to the results of a previous study, which reported that active coping, rather than passive coping, should be strengthened for effective stress coping [45]. According to Lazarus and Folkman [35], as a general trend, when a stressful situation is considered controllable, the stress can be dealt with through active actions, such as problem-solving, or through passive actions, such as denial and escape. In March 2022, when the participants of this study underwent the preceptorship, the medical staff could not avoid infection due to the spread of COVID-19 in Korea, and it is assumed that they had many difficulties in training new nurses because of a lack of workforce in the department. In the self-reflective journals written by preceptor nurses, it was confirmed that they struggled with the lack of workforce and organizational support due to the COVID-19 pandemic, but were forced to accept the situation. Therefore, they experienced stress and work overload, but exhibited passive coping strategies in a natural disaster setting where they could not actively solve the problem. A systematic review that studied healthcare workers’ coping behaviors during the COVID-19 pandemic also reported that, unlike previous studies, passive coping strategies reduced psychological distress by blocking unpleasant and negative emotions [46]. The average clinical experience of the participants of this study was 4.8 years, with most of the nurses having worked for <6 years. They had to train new nurses and perform heavy tasks at a time when their clinical experience was not abundant; hence, it is possible that they did not cope with stress properly [13]. However, considering the results of this study, which had not been proven to be significant, it is necessary to conduct a repetitive study on the effects of the preceptors’ active coping strategies by re-applying this program after the COVID-19 pandemic resolves.

Second, the PRPP of this study did not significantly reduce the burden on the preceptor nurses. Only 19.1% of the participants voluntarily performed the role of a preceptor, and 53.2% of the preceptors did it for the first time; hence, it can be thought that training new nurses while performing existing tasks might have been considered a burden. The participants of this study were often forced to start the preceptor role at the recommendation of a team leader, and it was confirmed that they felt burdened by the role of the preceptor when they were not prepared for it, as per the self-reflective journal. Owing to the situational characteristics of the COVID-19 pandemic, the study participants often trained two new nurses at once. As the new nurses’ time to work independently approached, the preceptors felt pressure and mainly expressed negative emotions. In addition, at the end of the preceptorship, it was found that new nurses were newly assigned to the department owing to the COVID-19 situation, and the burden of starting training again did not decrease. To reduce the burden of work for the preceptor nurses, organizational support of medical institutions is needed, administrative and nursing managers for preceptor nurses are important, and it is necessary to provide a basis for appropriate work sharing and role performance given to preceptor nurses. In addition, at the organizational and national levels, a reasonable compensation system to provide a support workforce and to the work of preceptor nurses should be established.

Third, the PRPP in this study significantly increased the preceptor nurses’ social support. It has been reported that social support is necessary as a coping mechanism to reduce psychological distress and promote positive emotions in healthcare providers [47]. The participants of this study wrote in a self-reflective journal while participating in the PRPP, and the clinical nurse educators provided emotional support, comments, advice, and played a mentoring role in the two workshops. In addition, it is assumed that the preceptor program provided by the clinical nurse educators was not limited to a one-time education, but rather continued to support the preceptor role, communication skills, and feedback [48], thus, helping overcome the difficulties and providing social support. The type and level of the support system may affect the preceptors’ confidence levels. Therefore, close cooperation in hospital organization and the clinical field is required [49]. For hospital administrators, forming tangible, emotional, and informational partnerships with nurses will increase social support, which would be a positive factor for the preceptor nurses to grow into professionals [50]. In Korea, the clinical nurse educator system was introduced in 2019 to reduce nurses’ turnover rate and secure a stable medical workforce. The clinical nurse educator system is still at a nascent stage, and various attempts and evaluations should be made to firm up the system; however, it should be activated for continuous emotional management of the preceptor nurses. This study provides basic data for identifying the role and effectiveness of clinical nurse educators. The clinical nurse educator system may be presented as a way for efficient workforce management.

Finally, the PRPP in this study was found to have a significant effect on increasing the emotional intelligence of preceptor nurses. Emotional intelligence increases self-directedness through emotion control and behavioral patterns, and the ability to control pleasant emotions induces positive thoughts, improves constructive thinking, and increases positive conversations and experiences [51]. In addition, it has been reported that video demonstrations of communication, peer assessment, reflective practice, and continuous feedback are effective educational strategies to improve emotional intelligence [52]. In the PRPP conducted in this study, it is thought that the communication role-play via video demonstration conducted in the workshops, and the preparation of the self-reflective journals, were effective in improving the emotional intelligence of preceptor nurses. In addition, allowing the preceptor nurses to write their experiences in self-reflective journals and reflecting on those experiences through two workshops, are considered effective strategies in turning those experiences into positive ones. These results are consistent with the findings of a study, which reported that the emotional intelligence scores were improved significantly after applying brief emotional intelligence intervention to nurses for 3 months [53]. In the self-reflective journal written by the preceptor nurses, it was very difficult to perform both existing work and new-nurse education; however, this was an opportunity for them to improve their professional career. Notably, the difficult experience helped turn on a positive direction through the reflective practice process. This PRPP may contribute to strengthening stress coping and preventing burnout by improving emotional intelligence through reflective training of the preceptor nurses.

While the existing research focused on strengthening educational capabilities for preceptors, the PRPP in this study was led by a clinical nurse educator to develop a program for reflective practice of preceptors, provide social support as well as to improve emotional intelligence for efficient and positive stress coping. In addition, it is of great significance that during the preceptorship period, through continuous education and reflective practice for preceptor nurses, efforts were made to conduct a new higher-quality education for the nurses [48].

Despite these strengths, this study has several limitations. First, there is a limit in verifying the effectiveness of this program with the design used before and after the single group and without setting the control group in the research design. Second, there is a limit to generalizing the study results because it targeted preceptor nurses at a tertiary hospital.

## 5. Conclusions

The PRPP was applied to preceptor nurses who performed preceptorship for 12 weeks, and the preliminary effects of the preceptorship were measured. Writing and verbal strategies were combined for reflective practice. A self-reflective journal was used, and a reflective workshop using verbal strategies was implemented to improve emotional intelligence and social support. At the end of the program, preliminary effects of significantly improving stress coping, social support, and emotional intelligence were confirmed. The negative experiences that the participants experienced while performing preceptorship changed in a positive direction, as they reflected on their own and shared experiences and emotions with fellow preceptor nurses. In addition, the positive feedback received from the clinical nurse educators worked effectively, suggesting that this program has a preliminary effect on improving the emotional intelligence of preceptor nurses. These findings may be used as a preliminary basis for the development of programs aiming to help nurses achieve efficient stress response and emotional intelligence improvement of preceptors. Based on the above research results, the authors suggest that specific intervention measures for national and organizational support should be presented to reduce the burden on preceptor nurses, and propose repetitive research and development of active stress coping reinforcement programs by sampling preceptor nurses from various medical institutions in the future.

## Figures and Tables

**Figure 1 ijerph-19-13755-f001:**
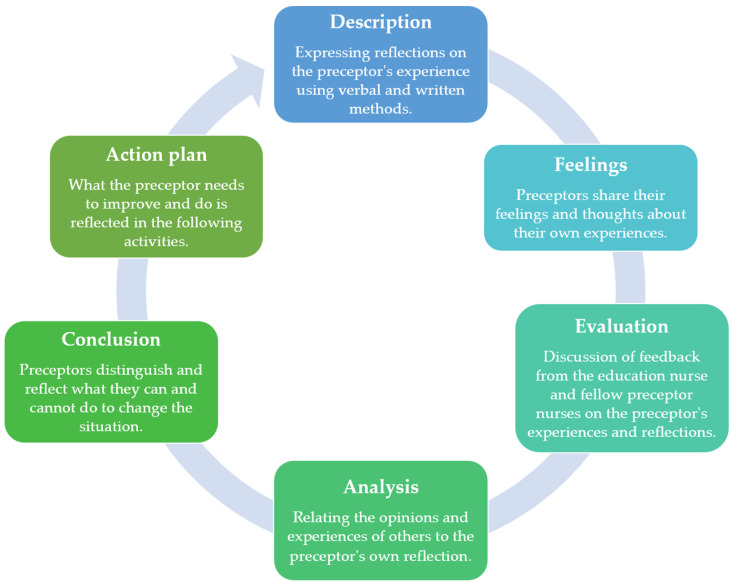
The theoretical framework of the Preceptor Reflective Practice Program (PRPP).

**Figure 2 ijerph-19-13755-f002:**
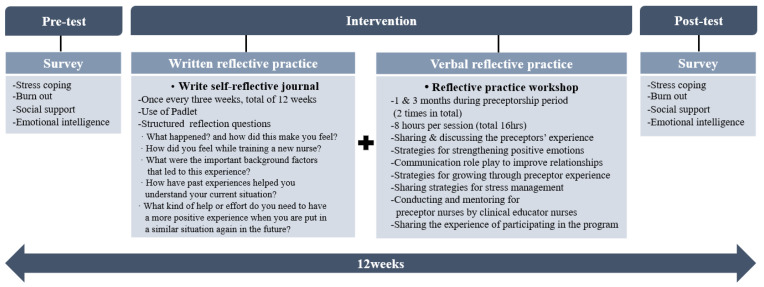
Preceptor Reflective Practice Program.

**Table 1 ijerph-19-13755-t001:** Contents of the Preceptor Reflective Practice Program activities for the preceptor nurses.

**The Components of the Written Reflective Practice**	
	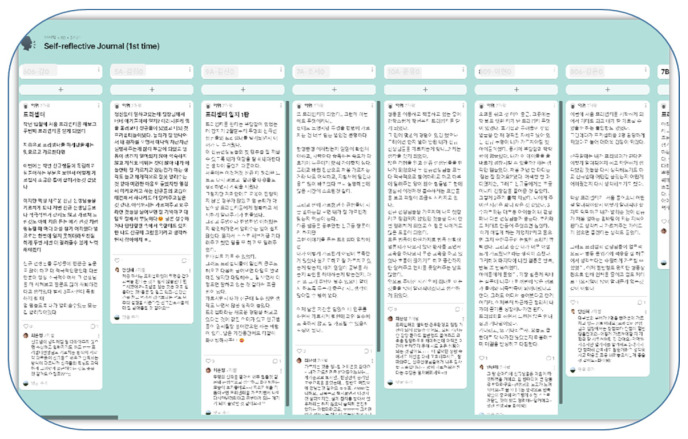
**The Components of the Verbal Reflective Practice**	
First workshop	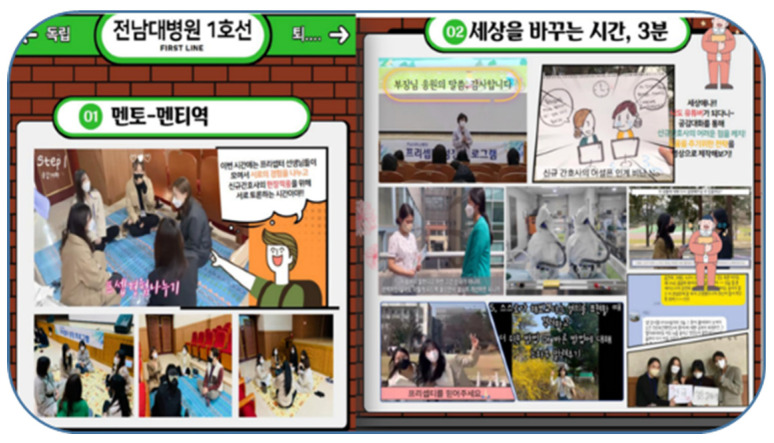
Second workshop	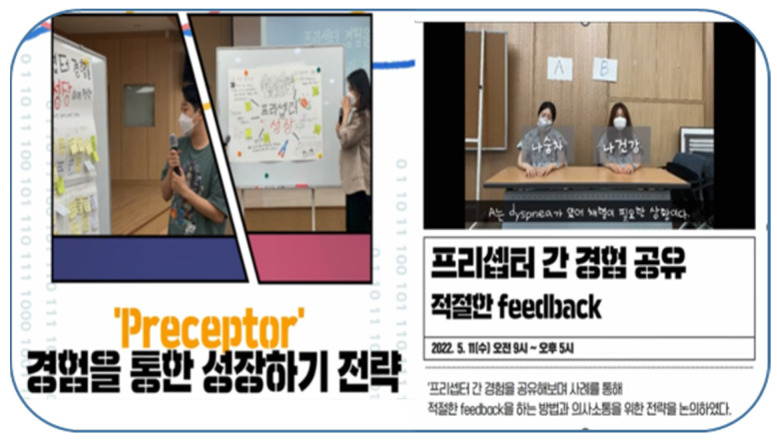

**Table 2 ijerph-19-13755-t002:** Participants’ general characteristics.

Characteristics	Categories	(N = 47)	
		*n* (%)	Mean (SD)
Age (years)	≤26	15 (31.9)	27.81 (2.52)
	27–29	25 (53.2)	
	≥30	7 (14.9)	
Sex	Men	3 (6.4)	
	Women	44 (93.6)	
Marital status	Single	44 (93.6)	
	Married	3 (6.4)	
Total working career (years)	<4	20 (42.6)	4.85 (2.80)
	4–6	20 (42.6)	
	>6	7 (14.9)	
Work unit	Medical ward	8 (17.0)	
	Surgical ward	10 (21.3)	
	Medical ICU	6 (12.8)	
	Surgical ICU	6 (12.8)	
	Other ICU	8 (17.0)	
	Others (OR, ER, Delivery room, Pediatric ward, Emergency ward)	9 (19.1)	
Voluntarily applied for the preceptor’s role	Yes	9 (19.1)	
	No	38 (80.9)	
Number of preceptor experiences (times)	1	25 (53.2)	
	2–5	22 (46.8)	
Whether to continue In the role of the preceptor	Yes	24 (51.1)	
	No	23 (48.9)	

ICU, Intensive Care Unit; OR, Operation Room; ER, Emergency Room.

**Table 3 ijerph-19-13755-t003:** Preliminary Effects of the preceptor stress management program on different variables.

Variables	(N = 47)			
	Pre-Test	Post-Test	t/z	*p*
	Mean (SD)	Mean (SD)		
Stress coping	2.80 (0.25)	2.89 (0.30)	−2.15	0.037
Active coping	3.01 (0.37)	3.06 (0.39)	−1.08	0.285
Problem-focused coping	2.88 (0.39)	2.97 (0.43)	−1.52	0.135
Seeking social support	3.14 (0.45)	3.16 (0.48)	−0.32	0.754
Passive coping	2.59 (0.29)	2.71 (0.34)	−2.52	0.015
Emotional focusing stress coping	2.32 (0.40)	2.49 (0.47)	−2.75	0.009
Desire thought	2.85 (0.40)	2.92 (0.34)	−1.14	0.258
Burden of preceptor	1.61 (0.57)	1.64 (0.73)	−0.35	0.725
Preceptor’s burden related to new nurses	1.52 (0.66)	1.54 (0.78)	−0.29	0.769 *
Preceptor’s burden related to oneself	1.63 (0.70)	1.63 (0.81)	−0.40	0.968
Preceptor’s burden related to colleagues or others	1.71 (0.70)	1.87 (0.83)	−1.30	0.201
Social support	3.23 (0.42)	3.36 (0.46)	−2.03	0.043 *
Emotional intelligence	3.52 (0.63)	3.82 (0.58)	−3.97	<0.001
Self-emotions appraisal	3.97 (0.73)	4.08 (0.60)	−1.17	0.242 *
Others-emotions appraisal	3.79 (0.73)	4.04 (0.65)	−2.84	0.007
Regulation of emotion	3.34 (0.84)	3.69 (0.71)	−2.92	0.005
Use of emotion	2.97 (0.77)	3.45 (0.89)	−3.59	<0.001 *

SD, standard deviation; * Wilcoxon signed-rank test.

## Data Availability

The data presented in this study are not publicly available because of privacy concerns.

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
