# Peer review of "Development and Preliminary Evaluation of the Effects of a Preceptor Reflective Practice Program: A Mixed-Method Research"

_ijerph, 2022, doi:10.3390/ijerph192113755_

Round 1

Reviewer 1 Report

qualitative analysis as the method must be more specific, that is, use one of the methods of qualitative research such as: phenomenology or qualitative descriptive method or ethnography or hermeneutics.

   also take into account a reference author of the qualitative studies for the interpretation of the data; In addition, the analysis must be done to avoid subjectivity through triangulation.

From the point of view of quantitative research, I consider that it is quite pertinent

Reviewer 2 Report

This manuscript is meaningful as a study on the development and implementation of an interesting program for preceptor nurses. However, overall, extensive correction of English expression is required, and there are unsatisfactory descriptions in the study design and results.

1. the theoretical framework of the development was not presented. Further literature review is needed on the effect of reflective practice on emotional management.

2. The program development process including validity test should be presented in detail.

3. The experimental design with control group is required to measure the effect of the developed program. The reason for choosing a mixed method design for one group should be clearly presented. The presented qualitative results have limitations in describing the effectiveness of the program. Therefore, in the conclusion of this manuscript, there is insufficient evidence to assert that the PRPP is effective.

Round 2

Reviewer 1 Report

can be published has made the suggested corrections

Author Response

Thank you for allowing me to publish my research.

Reviewer 2 Report

Most of the judges' suggestions have been reflected and revised.

However, the theoretical framework of intervention development and the stages of development have not been clearly presented.

Author Response

The authors would like to thank the reviewer for his/her constructive critique to improve the manuscript. We have made every effort to address the issues raised and to respond to your comments. The revisions are indicated in red font in the revised manuscript. Please find next, a detailed, point-by-point response to the reviewer's comments. We hope that our revisions will meet the reviewer’s expectations and that the revised manuscript is now suitable for publication.
